# Impact of Diquat on the Intestinal Health and the Composition and Function of the Gut Microbiome

**DOI:** 10.3390/antiox14060721

**Published:** 2025-06-12

**Authors:** Jiao He, Qing Tang, Yan-Cun Liu, Li-Jun Wang, Yan-Fen Chai

**Affiliations:** Department of Emergency Medicine, Tianjin Medical University General Hospital, 154 Anshan Road, Tianjin 300052, China; hejiao714@tmu.edu.cn (J.H.); tangqing202407@163.com (Q.T.); yancunliu@tmu.edu.cn (Y.-C.L.)

**Keywords:** diquat, oxidative stress, inflammation, mitophagy, intestinal microbiome, metabolic products, antioxidants

## Abstract

Diquat (DQ) is extensively utilized as a herbicide in farming, and its intake can result in serious systemic toxicity due to its induction of oxidative stress (OS) and disruption of intestinal homeostasis. The gastrointestinal tract is one of the first systems exposed to DQ, and damage to this system can influence the general health of the host. Our review summarizes the toxic effects of DQ on the intestinal barrier integrity, gut microbiome, and microbial metabolites (e.g., short-chain fatty acids [SCFAs], bile acids). By elucidating the mechanisms linking DQ-induced OS to gut dysbiosis, mitochondrial dysfunction, and inflammation, our work provides critical insights into novel therapeutic strategies, including probiotics, antioxidants (e.g., hydroxytyrosol, curcumin), and selenium nanoparticles. These findings address a pressing gap in understanding environmental toxin-related gut pathology and offer potential interventions to mitigate systemic oxidative damage.

## 1. Introduction

Pesticides are the primary reason for toxicosis-associated accidental deaths in China, which is the leading producer of pesticides, with the United States (US) and Argentina following. As paraquat (PQ) ingestion has a particularly high mortality, its sale and use have already been limited in many countries. Consequently, diquat (DQ) has become a widely used substitute for PQ [1]. DQ was first produced in 1955 by the Imperial Chemical Industries (ICI), a British company, which afterwards realized its value as a herbicide and introduced it to the market in 1958 [2,3,4]. DQ possesses a high binding appetency to soil and other organic substances and exhibits a high solubility in water [5]. Hence, DQ enters aquatic ecosystems easily through direct discharge from factories and agriculture, and does harm to humans, ducks, fish and other aquatic animals [6]. Its exposure approaches include the digestive tract, the respiratory tract, the eyes, and the skin [3,7], resulting in dysfunction of many organs such as the kidneys, brain, heart, and liver [1,8]. In recent years, the increasing cases of DQ toxicoses have attracted global attention, and DQ was classified as a potential carcinogen by the US Environmental Protection Agency (EPA) [9]. Until now, there is no specific antidote for DQ poisoning [10]. Existing studies show that DQ poisoning may be related to oxidative stress (OS), inflammatory response, and the induction of apoptosis [11,12]. Nevertheless, the specific mechanisms of DQ-induced injury are largely unclear.

As everyone knows, the main approach of pesticide exposure is the digestive tract [13,14]. Recently, the research emphasis has turned to how pesticide exposure affects the host intestinal flora, and evidence supports the possibility that DQ could disrupt the structure and function of the gut microbiome, resulting in host metabolic disorders [15,16]. The human gut, being the largest barrier organ by surface area, has a complicated barrier structure that serves as a selectively permeable shield, responsible not only for nutrient absorption but also for protection against detrimental external factors [17,18]. It is composed of four major layers: the microbiome in the intestinal lumen, responsible for decolonizing pathogenic bacteria from the gut (a biological barrier); the unstirred water layer, comprising digestive fluid and other substances released by the intestinal epithelial cells (IECs) to inhibit bacterial attachment (a chemical barrier); the mucosal layer (including antimicrobial substances generated by IECs and Paneth cells); and lined IECs (a mechanical barrier) that act as the main constituent of the intestinal wall linked through tight junctions (TJs) [17,19,20,21]. TJs (claudin, ZO-1 and occludin) are intercellular adhesion complexes that serve as gatekeepers of the paracellular space [22].

The gut microbiota, a varied ecosystem, lives within the digestive tract and contains at least 1013–1014 microbial cells and more than 2000 diverse species, involving bacteria, viruses, fungi and parasites [23,24]. The primary function of gut microbiota is to regulate the host metabolism as a ‘central metabolic organ’ and defend against harmful stimuli and pathogens [25]. The gut microbiota is crucial for preserving the integrity of IECs and beneficial for maintaining the host’s defense system homeostasis through generating antimicrobial compounds, facilitating mucus secretion, and inhibiting the colonization of pathogens [26]. A dynamic but relatively stable gut microbiota contributes to human health, while dysbiosis may lead to inflammatory responses and various diseases such as colon cancers, obesity, and diabetes [27,28]. Figuring out the relationship between DQ and the functionality of the intestinal microbiota can provide a better understanding of DQ-induced toxicity and new therapeutic strategies in DQ-related damage.

This review summarizes the potential detrimental effect that DQ has on the intestine, with special attention paid to the intestinal microbiota. Firstly, we expound on impact of DQ’s inherent toxicity on the gut. Next, we discuss its effect on the intestinal microbiome and its metabolites.

## 2. Toxic Effects of DQ on the Intestine and the Gut Barrier

DQ crosses the cell membrane primarily through diffusion and, to a lesser degree, by active transport through cation pumps [3]. DQ presents a direct toxic effect on the IECs and firstly makes contact with the intestinal mucosa, resulting in extensive damage. Meanwhile, the release of a large number of inflammatory factors can directly or indirectly lead to intestinal damage, impairing the intestinal barrier, consequently accelerating DQ absorption. The TJ between IECs is one of the essential structural bases for preserving the integrity of the gut barrier [29]. Accumulating evidence has demonstrated that OS destroys the TJs in many ways, leading to intestinal epithelial barrier dysfunction [30,31]. According to research by Tian et al. (2023), OS induced by DQ results in morphological abnormalities and functional damage in the small intestine [32].

### 2.1. Oxidative Stress and Inflammation

A survey of existing studies on pesticides showed that all pesticides can cause OS [33], which involves an imbalance between the production of reactive oxygen species (ROS) and the capability of the defense system to eliminate the superfluous ROS [34,35,36]. Normally, there is an actional equilibrium between the oxidants and antioxidants in biological systems [37]. Excessive ROS can damage cellular proteins, DNA, and lipids, causing fatal cellular injury [38]. The gut sits at the interface between the organism and its lumen environment [39], serving as a barrier against invading intestinal pathogens, and thus is more susceptible to oxidative injury [40,41,42,43,44]. The gut barrier consists of monolayer cells and is the main site of OS response [45,46]. OS mainly injures intestinal health by inducing cell injury, inflammation and gut dysfunction [47,48]. Some studies have demonstrated that the overproduction of ROS in the intestine can disrupt the epithelium function, increase the permeability of the gut, damage IECs and disturb nutrient absorption [44,49,50]. For example, studies have reported that some oxidants augment the permeability of the gut barrier through disturbing TJs, accompanied by the disruption of the nuclear factor kappa-B (NF-κB) signal pathway and inflammation [46,51]. It has been reported that OS interacts with inflammation, ROS contributes to inflammation, and in turn, inflammation contributes to the production of ROS [52,53,54,55]. Studies have found that OS can activate the NF-κB signal pathway, which in turn facilitates the release of proinflammatory cytokines [56,57]. In addition, extensive studies have shown that OS can disrupt the natural structure of the small gut, involving increasing crypt depth (CD) and decreasing villus height (VH) and the ratio of VH to CD (V/C), which can be regarded as an indicator of the absorption ability [58,59,60]. Integrating these results, we speculate that therapy to reduce oxidative injury and inflammatory response may be beneficial in maintaining the function of the epithelial barrier, as well as intestinal and host health.

Moreover, OS is regarded as a key factor in disturbing the balance of the intestinal microbiome and notably reducing the microbiota diversity [61,62]. Recently, OS has been reported to influence the structure as well as the functionality of the gut microbiota [63] and change the metabolites [64]. Furthermore, the metabolites of the microbiome are also crucial for intestinal microbiome-mediated redox adjustment [65,66]. Some metabolites, including short-chain fatty acids (SCFAs), polyphenolic and tryptophan derivatives, lipopolysaccharides (LPS), and trimethylamine-N-oxide (TMAO), were found to modulate OS [67,68,69,70]. Research has found that ingesting probiotics alone or dietary supplementation with probiotics could alleviate OS and change the activities of pivotal antioxidant enzymes [71]. These findings can offer novel insights and strategies for the therapy of intestinal OS illnesses.

### 2.2. The Role of OS and Inflammation in DQ-Induced Intestinal Injury

DQ can produce superoxide anion radicals by the molecular oxygen, which breaks the redox equilibrium of the gut, leading to OS and inflammation [72,73,74,75]. The ability to induce OS is greater than that of other herbicides due to DQ’s high value of redox potential [76]. Extensive studies have shown that OS ruins the structure of the gut and results in apoptosis in enterocytes [77,78]. It was also reported that OS can induce essential mitochondrial channel opening and activate the apoptosis pathway, which depends on mitochondria, causing cell death [79,80].

The main antioxidant defense machineries for humans consist of antioxidants and antioxidant enzymes, involving superoxide dismutase (SOD), catalase (CAT) and glutathione peroxidase (GSH-Px) [81]. DQ was reported to elevate the content of malondialdehyde (MDA) and reduce CAT, SOD and GSH-Px in jejunal mucosa [82,83,84,85]. In addition, studies have found that SOD can effectively boost antioxidant ability, maintain mucosal barrier integrity, and inhibit pro-inflammatory responses [86,87]. Li et al. obtained a novel *SOD* gene from the *Hydrogenobacter thermophilus* strain (Ht), and their study demonstrated that HtSOD mitigates OS and intestinal injury by decreasing total ROS in DQ-treated mice [42].

Extensive evidence has revealed that the nuclear factor erythroid 2-related factor 2 (Nrf2) signal pathway and mitophagy can mitigate OS. It was reported that Nrf2 primarily adjusts the transcription of antioxidant genes, including *CAT*, *SOD*, and *heme oxygenase-1* (*HO-1*), by regulating the antioxidant defense system [88,89,90]. Mechanistic research in IPEC-J2 cells exposed to DQ demonstrated that hydroxytyrosol (HT) alleviated intestinal oxidative impairment through activating the Nrf2 signal pathway and facilitating mitophagy. HT, a natural polyphenolic substance, mainly exists in olives and presents potent antioxidant capacity [91]. DQ induced OS and damaged intestinal barrier function, whereas HT mitigated these adverse phenomena. This research showed that targeting the Nrf2 signaling pathway and mitophagy may become a hopeful tactic for curing OS-related damage induced by DQ [92]. For instance, recent advances in selenium research revealed that dietary selenium nanoparticles (SeNPs) triggered the Nrf2 pathway to suppress the activation of inflammasome caused by OS in mice treated with DQ, consequently alleviating OS-related intestinal barrier dysfunction [81,93].

KEGG analysis of an original research showed that differentially expressed genes (DEGs) in mice after exposure to DQ were primarily enriched in the NF-κB signaling pathway [81]. Evidence supported that Nrf2 can inhibit the activation of NLRP3 inflammasomes, which are crucial for the processing and release of inflammatory cytokines, including IL-6, IL-18, IL-1β and TNF-α [94]. Qiao et al. (2022) demonstrated that DQ exposure increased the level of the NLRP3 inflammasome and subsequently caused an increase in IL-18 and IL-1β in mice [95]. Another study has shown that the concentration of IL-6 and TNF-α is significantly elevated in the piglet model of DQ-induced OS [96]. TNF-α, a proinflammatory cytokine, plays an important role in systemic inflammation. The collected data confirmed that TNF-α can result in an increase in intestinal epithelial TJ permeability [97,98].

As mentioned before, due to its characteristic biological structure and functional location, the gut is especially vulnerable to DQ, which causes severe OS. Thus, regulating OS has become a significant means to ameliorate the damage induced by DQ. Measures such as developing antioxidants and understanding their mechanisms and pathways can help alleviate OS induced by DQ and reduce the risk of intestinal damage.

### 2.3. Mitophagy

DQ is also identified as leading to mitochondrial damage. Studies have revealed that mitochondrial dysfunction is involved in DQ poisoning [12,82,99]. IECs contain abundant mitochondria, which are the main production place and the primary target of ROS [82,100,101,102]. Once exposed to external noxious stimuli, the impaired mitochondria produce excessive ROS, which further worsens oxidative damage [103,104,105]. Chen et al. (2021) [106] found that DQ exposure led to ROS accumulation, which destroyed mitochondrial DNA (mtDNA) and mitochondrial enzymes and caused mitochondrial dysfunction in the jejunum of weanling piglets. Such damaged mitochondrial function, in turn, exacerbated the production of mtROS, creating a vicious cycle. Moreover, research has shown that the accumulation of mtROS reduces the microbial diversity of the gut as well as antimicrobial defenses [107,108]. In this regard, alleviating mitochondrial disorders may be a promising strategy for the improvement of OS-related damage caused by DQ.

Previous investigations have demonstrated that OS can trigger mitophagy in DQ-induced intestinal damage [109]. It has been reported that DQ-induced OS could impair the function of the epithelium in piglets, with the mitochondrial disorder of the jejunal mucosa and mitophagy [83,110]. Mitophagy can eliminate excess ROS through phagocytosing impaired mitochondria and degrading them [111,112,113]. It was reported that OS can trigger mitophagy through the PINK1-Parkin signaling pathway [114,115]. In existing studies, researchers found that DQ exposure induces the production of autophagic vesicles and augments mitophagy-associated proteins, indicating that the mitophagy pathway has been initiated [92]. Recent studies have indicated that mitophagy plays a protective role in preventing cellular death triggered by OS in DQ-treated piglets. This process involves the selective engulfment of impaired mitochondria by autophagosomes, followed by their subsequent breakdown within lysosomes. Such cellular machinery not only maintains intestinal epithelial function but also decreases additional oxidative injury [82].

Recently, a growing body of studies have indicated that metabolites derived from gut microbiota play a significant role in modulating OS through microbial activity in the intestinal environment. SCFAs, especially acetate, propionate, and butyrate, are metabolic byproducts generated by beneficial gut microbiota during the breakdown and fermentation of dietary fiber [116,117,118]. These SCFAs serve as crucial energy substrates for IECs, facilitating their growth and differentiation. Additionally, they play a significant role in modulating gut functionality, immune responses, and preserving the homeostasis of microbial communities. In addition, SCFAs have been confirmed to possess antioxidant abilities, and among them, butyrate has received particular attention [119]. Extensive studies conducted both in animal models and cell cultures have verified that butyrate possesses the ability to effectively alleviate OS within the intestinal environment [120,121,122,123,124]. A study has shown that butyrate serves as a signal molecule safeguarding the liver from OS through stimulating the Nrf2 signal pathway in rats [124]. Furthermore, butyrate also possesses numerous functions, involving providing the major energy for intestinal cells, promoting TJ protein formation [119,125], suppressing the proliferation of pathogens, contributing to the growth of enterocytes, and restraining the intestinal inflammation [126,127,128]. In addition, some researchers indicated that butyrate has the capacity to restore mitochondrial energy deficiencies [129]. Data from Wang et al. (2019) [55] indicated that DQ injection caused serious intestinal OS in pigs, while butyrate relieved intestinal OS and inflammation and improved mitochondrial function through selectively inducing mitophagy. Wang’s research revealed that butyrate notably elevated protein levels of PINK1 and Parkin in mitochondria. Moreover, Lee et al. (2012) found that butyrate increased the survival of hamster ovary cells through recruiting Parkin and inducing mitophagy to eliminate damaged mitochondria [130]. Furthermore, previous studies have demonstrated that elevated butyrate concentrations upregulated mitophagy-related genes in cell lines from autistic boys under OS [131].

As previously mentioned, Nrf2 reduces OS through regulating the transcription of genes involved in antioxidant defense mechanisms, whereas mitophagy eliminates impaired mitochondria at the source. The synergistic effect of these two processes contributes to preserving cellular stability and improving the impairments induced by OS. Nevertheless, the precise interplay between Nrf2 and mitophagy is not yet entirely clear and needs further investigation.

The molecular mechanisms mentioned above are outlined in Figure 1.

### 2.4. Impact of DQ on Intestinal Barrier Integrity

It is generally acknowledged that the gut barrier, as the main shield against the harsh conditions within the lumen of the intestine, is critical for safeguarding the organism against toxins, antigens, and pathogens [132,133,134]. The intestinal epithelium comprises enterocytes, Paneth cells, and goblet cells, collectively constituting the principal cellular constituents of the intestinal barrier system that orchestrates mucosal defense mechanisms [135]. Goblet cells, a significant component of IECs, are the primary source of mucin secretion. In the gut, *mucin 2* (*MUC2*) is the predominant mucin secreted by goblet cells [136] and serves as a key regulatory gene for the intestinal mucus layer, playing a significant role in maintaining intestinal barrier integrity [137]. Research has found that DQ exposure decreases the number of goblet cells, thereby diminishing MUC2 secretion. The deficiency of MUC2 compromises the integrity of the inner mucus layer, permitting pathogenic bacterial–epithelial interactions that trigger innate immune activation and chronic mucosal inflammation [138]. Animal studies have revealed that MUC2-deficient mice have chronic colonic inflammation [139].

Studies have demonstrated that DQ impaired the integrity of the gut barrier in mice, leading to fragmented brush borders and a disarrayed epithelium [83]. The excessive ROS in the intestinal epithelium can elevate intestinal permeability, enabling the translocation of detrimental luminal components into the bloodstream (as described in Figure 2) and elevating the likelihood of systemic inflammatory responses [140]. Gut integrity plays a significant role in evaluating gut health and could be quantified by various measurable parameters, such as diamine oxidase (DAO), D-lactate (DLA), VH, and CD [141]. DAO, an enzyme found within cells, is primarily produced and distributed in the intestinal epithelium [142]. DLA is a special byproduct generated by microbial metabolic processes. Increased serum concentrations of DAO and DLA commonly suggest gut barrier injury [143,144]. Chen et al. (2022) demonstrated that DQ-treated pigs showed elevated serum DAO and DLA levels, verifying the impairment of gut barrier integrity [145].

VH and CD serve as direct morphological parameters for assessing the structural integrity of the gut mucosa [146]. TJs form the fundamental building blocks of the gut barrier, with occludin being the earliest discovered TJ protein, while ZO-1 and claudin play critical roles in maintaining the physiological and structural integrity of the paracellular barrier [147]. It has been reported that claudins are crucial components of TJs and determine their barrier properties and paracellular permeability [148]. Both animal and cell culture studies showed that DQ-related OS compromises the integrity of IECs, evidenced by the disruption of TJs and the reduced viability of IECs, ultimately leading to the dysfunction of the nutrient metabolism [149,150,151]. Additionally, Wen et al. (2020) found that DQ negatively impacted the morphological development of the small intestine, decreasing VH and the V/C ratio, while also downregulating the expression of ZO-1 and occludin proteins in piglet intestinal tissue [152].

Furthermore, research has found that the atrophy of intestinal villi and barrier dysfunction caused by OS were partially related to the programmed cell death of IECs [153]. It was found that the impairment of the gut barrier is always accompanied by the apoptosis of IECs [154,155,156]. As everyone knows, apoptosis is modulated by various molecular mechanisms, with significant involvement from the Bcl-2 and caspase protein families [157]. Chen et al. (2022) revealed that DQ exposure obviously increased the expression levels of caspase-9, caspase-3, and Bax in the mucosal layers of the duodenum and jejunum, indicating that DQ induced apoptosis of IECs [145]. Furthermore, studies showed that DQ exposure led to atrophied intestinal villi, featuring fragmented intestinal brush borders and disordered epithelial layers, resulting from the apoptosis and shedding of IECs [21,158].

## 3. Effects of DQ on the Composition and Function of the Intestinal Microbiome

Increasing evidence has confirmed the crucial function of the intestinal microbiome in the physiological and pathological states of the body. Studies have revealed that the intestinal microbiome helps repair intestinal mucosal barrier injury [159] and also plays vital roles in modulating host OS and inflammation [160,161,162]. It has been reported that DQ reduced gut microbiota diversity, whereas HtSOD alleviated the decrease in microbial diversity caused by DQ and contributed to the growth of beneficial microbiota [42]. The distinct gut microbial profiles between DQ-exposed animals and the normal group have been exhibited in several studies.

Yuan et al. (2024) [81] found altered ileal mucosal microbiota composition in DQ-treated piglets. In comparison with normal piglets, DQ exposure reduced *Acidobacteria* populations while elevating the levels of *Clostridium* and *Turicibacter*. *Clostridium* genera are Gram-positive obligate anaerobic bacteria and are usually deemed as pathogenic microorganisms [163], while *Turicibacter* might influence gut health and contribute to the development of multiple disorders, such as diabetes and inflammation [164]. The elevated levels of *Clostridium* and *Turicibacter* showed exacerbated ileal inflammation resulting from DQ exposure. 16S rRNA analyses in the study by Wu et al. (2022) suggested that DQ administration disrupted the microbial balance in rats, markedly reducing *Firmicutes* and *Lactobacillus* populations while augmenting *Proteobacteria* abundance [165]. This is an inconsistent result to that of Yuan et al., which we hypothesize may be due to interspecies differences. *Firmicutes*, a Gram-positive bacterial phylum, constitutes the predominant microbial group in healthy human gut microbiota [17,166]. It encompasses various beneficial bacterial species capable of metabolizing SCFA salts, including acetate and lactate, thus influencing the balance of symbiotic microorganisms in the intestine, alleviating gut inflammation, and maintaining immune homeostasis [167,168,169]. *Proteobacteria* phylum contains the majority of conditional pathogens, such as *Helicobacter*, *Vibrio*, *Salmonella* and *Escherichia* [170]. There is a broad consensus that numerous bacterial species within the *Proteobacteria* phylum can induce persistent gut inflammation and tissue damage in piglets [171,172]. STAMP analyses in this study showed notably elevated levels of conditional pathogens *Escherichia coli* and *Proteobacteria*, alongside reduced populations of potentially beneficial microbes, including *Firmicutes*, *Lactobacillus*, and *Akkermansia*, in DQ-treated rats. Pearson’s correlation analysis revealed that beneficial microorganisms, including *Butyricicoccus* and *Faecalibacterium*, showed inverse relationships with MDA levels, further verifying the link between microbial community changes and DQ-induced OS [165]. Han et al. (2023) demonstrated that DQ exposure decreased *Firmicutes* populations, consistent with previous research findings, while simultaneously increasing *Bacteroidetes* levels in mice [173]. Recent research has indicated that the *Firmicutes*/*Bacteroidetes* (F/B) ratio is positively related to the antioxidant ability in piglets [174] and is recognized as a significant indicator of the health of the gut microbiota [175]. Furthermore, an article reported that DQ exposure increased *Parabacteroides* populations in mice, consistent with another study exhibiting a positive correlation between *Parabacteroides* abundance and OS markers in murine models [176]. The collected data confirmed that *Parabacteroides* is a core member of the intestinal microbiome and is related to the development of OS and inflammation [177]. Yang et al. (2024) found that DQ markedly decreased *Lactobacillus* and *Alistipes* populations in broilers [178]. *Alistipes* serves as a key contributor to SCFAs production, including acetate and propionate [179]. Research has indicated that *Alistipes* potentially exerts protective influences against certain pathological conditions, involving colitis, cardiovascular diseases, and tumor immunotherapy [180]. This further verifies that DQ can reduce the populations of advantageous microbial species, thus impairing gut homeostasis. The results of Fu et al. (2021) indicated that DQ exposure increased the populations of *Ruminococcaceae* UCG-005 and *Eubacterium coprostanoligenes* in weaned piglets [171]. A previous study demonstrated that *Ruminococcaceae* UCG-005 is a stable constituent of the intestinal microbiome in piglets [181]. Research has revealed that *Ruminococcaceae* UCG-005 is positively related to metabolic diseases and chronic inflammation in weaned piglets [182]. Furthermore, another study has indicated that the increase in *Ruminococcaceae* UCG-005 in piglets exposed to DQ implies the exacerbation of the gut milieu because the population levels of *Ruminococcus* within the gastrointestinal system significantly influence the occurrence of diarrhea [183,184]. *Eubacterium coprostanoligenes* is widely recognized for its capacity to transform cholesterol into coprostanol, thereby decreasing circulating cholesterol levels. Some findings have suggested that *Eubacterium coprostanoligenes* populations are elevated in correlation with rising serum cholesterol concentrations in piglets exposed to thermal stress [185]. Based on the data acquired, we hypothesize that OS induced an elevation in circulating cholesterol levels, subsequently elevating the abundance of *Eubacterium coprostanoligenes*.

Recent studies have indicated that there exists a connection between mitochondria and the intestinal microbiome. On the one hand, the intestinal microbiome has been found to modulate pivotal transcriptional factors that play crucial roles in the process of mitochondrial biogenesis. In addition, the intestinal microbiome and its byproducts, including SCFAs and secondary bile acids, also play a stimulative role in energy generation, OS, and inflammation reduction through diminishing TNF-α-driven responses and inflammasomes like NLRP3 [95]. Furthermore, mitochondrial events, especially mtROS generation, are crucial for modulating the intestinal microbiome through regulating the functionality of the gut barrier [186,187]. One study has shown that selenium deprivation disrupted redox homeostasis and altered the intestinal microbiome composition, which is more vulnerable to gut barrier impairment upon DQ exposure. This effect is mediated via the Nrf2-dependent regulation of NLRP3 inflammasome signaling in murine models subjected to DQ stimulation. Nutritional intervention using biologically synthesized SeNPs produced by *Lactobacillus casei* ATCC 393 significantly improved the impairment of the gut barrier through increasing the antioxidative ability, ameliorating mitochondrial integrity and functionality, and preserving intestinal ecological balance through the Nrf2-regulated NLRP3 signal cascade [95].

This field would be worth investigating further, as although there is information available sketching the effects of DQ on the composition and functionality of intestinal microbiota, the exact mechanisms remain challenging to completely characterize owing to interspecies and individual differences, necessitating further comprehensive research. Various lifestyle determinants contribute to the unique composition of individual intestinal microbiota, including dietary habits, probiotic consumption, age, environmental exposures, and exercise regimens [188,189].

In conclusion, DQ exerts toxic effects on intestinal microbial communities through modifying their compositional profile and functional characteristics. These microbial changes may provide valuable biomarker potential for evaluating DQ-induced toxicity.

## 4. Impact of DQ on Metabolites

Microbial-derived metabolites, including SCFAs, tryptophan metabolites, and bile acids (BAs), have been demonstrated to significantly influence gut homeostasis [190]. These metabolites from different microorganisms have been shown to regulate intestinal physiology and influence metabolic processes across multiple organ systems, including the liver, muscle, and brain [191,192,193]. They possess dual regulatory capabilities, facilitating the development and activity of immunoregulatory cells while simultaneously restraining pro-inflammatory pathways, thereby preserving both intestinal and systemic physiological balance in the host [194]. BAs are synthesized through cholesterol catabolism, undergoing hepatic conversion to primary forms (PBAs) before intestinal transit, where microbial biotransformation generates secondary metabolites (SBAs), preserving physiological balance via enterohepatic recirculation [195]. Studies in humans and mice have indicated that BAs are involved in modulating intestinal inflammation, tumorigenesis, and immune function [196,197]. SBAs undergo microbial-mediated conversion to generate TUDCA, a bioactive compound shown to stimulate Nrf2 signaling, increase antioxidative enzyme expression, and exert antioxidative effects [195,198]. Thereby, the regulation of intestinal microbial communities and their metabolic byproducts emerges as a promising therapeutic approach for ameliorating OS in the gastrointestinal tract. It was reported that polyphenolic substances exert beneficial effects on microbial ecosystem stability and normal metabolism [199,200]. As previously mentioned, HT is a natural polyphenolic compound and possesses potent antioxidant ability [201]. Wen et al. (2024) [137] reported that DQ caused disorders in BA metabolism, including lower levels of PBAs, hyocholic acid (HCA), hyodeoxycholic acid (HDCA), and TUDCA, while treatment with HT partially reversed these alterations. This study also revealed a positive association among the expression levels of HCA, Nrf2, and CAT, suggesting that HCA may possess antioxidant properties. Furthermore, HDCA was found to enhance the activity of antioxidative enzymes and then alleviate oxidative damage and inflammation. Moreover, some studies have revealed that HDCA has the potential to serve as an intrinsic regulator that inhibits inflammatory signaling mechanisms [202]. Accumulating evidence has demonstrated that TUDCA is capable of modulating the activity of antioxidative enzymes, which helps reduce OS [203,204,205]. This perspective was further supported by a positive association observed among the expression levels of TUDCA, Nrf2 and CAT. It is widely recognized that butyrate-producing bacteria are predominantly classified within the *Firmicutes* phylum, among which *Faecalibacterium prausnitzii* and *Roseburia* are acknowledged as the most potent butyrate producers. Notably, *Faecalibacterium prausnitzii* serves as a principal contributor to colonic butyrate levels due to its high relative abundance and efficient butyrogenic pathways [206]. In another study that also used HT as an intervention, researchers reported that HT administration counteracts the DQ-induced decrease in the relative abundance of *Firmicutes* and butyrate levels in mice, implying a potential connection between butyrate and the antioxidant ability enhanced by HT [173].

Studies have revealed that DQ decreases the production of microbial tryptophan, thereby decreasing its availability in the host [207]. Tryptophan and its metabolites are recognized for their function as signaling molecules that facilitate communication between the intestinal microbiome and host cells [208]. Research has found that OS triggered by DQ may increase tryptophan metabolism in pigs [84,207], and this metabolic shift drives indoxyl sulfate (IS) overproduction. IS, a uremic toxin, builds up in plasma as kidney function declines, accelerating the progression of chronic kidney disease (CKD) [209]. In some situations, IS can be beneficial to the host, most markedly by inhibiting inflammation [210] and enhancing gut barrier function [211]. Nevertheless, in patients with CKD, impaired kidney function results in a significant elevation of IS levels in the plasma, inducing detrimental effects [212]. Tryptophanases produced by gut bacteria transform tryptophan into indole, which is subsequently absorbed and transformed by the host into IS. TDO, the hepatic rate-limiting enzyme governing systemic tryptophan catabolism, mediates the oxidative cleavage of tryptophan into kynurenine pathway metabolites [213,214]. Previous studies have demonstrated that DQ exposure can elevate the TDO mRNA level in the liver of piglets [84]. Researchers speculated that the observed decrease in circulating tryptophan levels may be mechanistically linked to the DQ-induced upregulation of hepatic TDO activity [207]. The collected data confirmed that *Bacteroides* species predominantly harbor the most abundant tryptophanases in the intestines of most individuals. Research has found that DQ exposure increased the abundance of *Bacteroides* [63]. Researchers identified a widely distributed family of tryptophanases in gut-commensal *Bacteroides* and found that the targeted ablation of this gene eliminated the generation of indole in vitro [209]. This gives us some new insights: modulating the gut microbiota to reduce the indole concentration, thus decreasing IS in the circulation, may offer a novel strategy to lower the detrimental effects of kidney injury caused by DQ (the gut–kidney axis). Studies have revealed that most human-related bacterial species capable of generating indole are typically low-abundance colonizers of the intestine or pathogens [215,216]. This information may help us better comprehend, forecast, and reprogram IS levels in vivo. For instance, researchers demonstrated that through deleting or changing a single gene and its orthologs, they could clear or obviously decrease the content of urinary IS. In addition, it was also observed that rationally the changing diet could contribute to the proliferation of a tryptophanase-negative *Bacteroides* species into a model intestinal community, proving the feasibility of utilizing dietary interventions to reduce the microbiota’s ability to generate indole [209].

In the research by Fu et al. (2021) [171], the effects of DQ on the intestinal metabolic profiles was investigated. 3-methyldioxyindole is an oxidative metabolite of 3-methylindole, generated by colonic bacteria, and it plays a role in tryptophan metabolism. In Fu’s results, a notable reduction in 3-methyldioxyindole was found in DQ-treated weaned piglets. Correlation analysis helped identify several bacterial genera that may play a role in host metabolism. The levels of *Acidaminobacter* showed a negative relationship with two metabolites and a positive relationship with eight metabolites, while *Terrisporobacter* exhibited a positive relationship with one metabolite and a negative relationship with six metabolites. Integrating these results, researchers suggested that the destruction of intestinal microbial structure and metabolic balance is likely the primary factor contributing to the reduction in the antioxidative ability of DQ-exposed piglets.

Studies have found that the intestinal microbiome also affects host health by influencing the host metabolome [217,218]. Modifications in the structure of the intestinal microbiome can alleviate host OS through altering serum metabolomics. It was reported that HT administration increased the levels of multiple serum metabolites that help mitigate OS [173]. As an example, 20-carboxy arachidonic acid undergoes metabolism through cytochrome P450 omega-oxidase, generating 20-hydroxyeicosatetraenoic acid, a metabolite with demonstrated antioxidative and anti-inflammatory activities [219]. Likewise, 3-hydroxytetradecanoyl carnitine plays a role in mitochondrial activity and oxidation [220]. Malic acid serves as a crucial intermediate metabolite of the tricarboxylic acid cycle (TCA). Evidence has demonstrated that it increased the antioxidative ability in the liver of fish [221].

## 5. Several Therapies to Alleviate the Toxic Effects of DQ

Out of the current treatment strategies, probiotics have obtained the most attention [222]. Studies have revealed that probiotics can exhibit the antioxidative capacity via diverse approaches, including upregulating the antioxidases in the host and modulating intestinal microbiota [223]. As is well-known to all, *lactic acid bacteria* (*LAB*) possess antioxidative and immunoregulatory abilities and are regarded as ideal probiotics [224]. It was reported that *LAB* could be utilized as a supplement to regulate Nrf2 and NF-κB signal pathways, and ameliorate intestinal inflammation [225]. The dominant genus among *LAB* is *Lactobacillus* [189]. It was reported that the genus *Lactobacillus* is positively related to serum total antioxidant [63,226] and can suppress ROS generation through restraining the activities of critical enzymes like NADPH oxidase and increase the ROS elimination capacity of the antioxidative system through modulating Nrf2 and NF-κB signaling pathways [227]. Research has shown that *Lactobacillus* can suppress the expression of TNF-α by the NF-κB pathway to relieve intestinal inflammation [225]. Feng et al. (2023) found that *Lactobacillus*-derived extracellular vesicles (EVs) [228,229] obtained from piglets could elevate antioxidant activity through ameliorating the gut barrier and reorganizing gut microbiota in mice after exposure to DQ [230]. *LAB*-EVs reshaped the gut microbiota through reducing the growth of detrimental bacteria like *Enterococcus* while promoting the growth of beneficial bacteria, including *Parasutterella* and *Erysipelatoclostridium*. As a result, these alterations ameliorated the intestinal barrier function, as demonstrated by elongated intestinal villi; elevated the activities of the antioxidative enzymes; and alleviated the oxidative injury. Spearman’s correlation analysis exhibited consistent results: *Erysipelotrichales* and *Parasutterella* showed a positive relationship with the antioxidative enzyme activities and a negative relationship with oxidative injury. Another study has shown that the inclusion of *Lactiplantibacillus plantarum* P8 in the diet facilitated an amelioration of the antioxidative ability, microstructure, and barrier integrity in the jejunal mucosa and a reduction in apoptosis in DQ-treated piglets. Transcriptome analysis of the jejunum further revealed that the positive effects of P8 might be correlated with the modulation of the NF-κB signal pathway [231].

The gut plays a significant role in preserving an organism’s arginine balance and arginine absorption, endogenous synthesis, and metabolism [232]. It was reported that OS reduced the content of serum arginine in piglets [85], and supplementation with arginine can diminish OS through elevating the total antioxidative ability in piglets [233]. Furthermore, another study has shown that arginine could maintain barrier function and decrease bacterial translocation in the gut of mice [234]. Research has demonstrated that supplementation with arginine notably decreased intestinal CD and TNF-α mRNA levels in jejunum exposed to DQ. This study supports that arginine supplementation in the diet promotes the gut health via ameliorating the intestinal morphology, modulating arginine availability and lowering inflammatory cytokine levels [149].

Resveratrol, a polyphenolic phytoalexin derived from plants like peanuts, has acquired significant attention for its antioxidative and anti-apoptotic properties. Recent studies have indicated that the administration of resveratrol as a dietary supplement elevated the antioxidative capacity in the host [235,236]. Additionally, resveratrol possesses a potent inhibitory effect on ROS generation and exhibits a wide range of biological activities, including anti-inflammatory, anti-obesity, and anti-aging properties [237,238]. Apigenin, a naturally occurring phytochemical flavonoid, is found in various plant-based foods, such as fruits and vegetables [239]. Recent studies have indicated that apigenin exhibits potential antioxidative, anti-apoptotic, and anti-inflammatory properties [239,240,241,242]. Zhou et al. (2022) [243] found that resveratrol and apigenin could reduce the MDA level and increase SOD and GSH-PX levels in DQ-treated pullets. Furthermore, resveratrol upregulated the levels of HO-1 mRNA in ileac and jejunal tissue, while apigenin increased NRF2 and HO-1. Both resveratrol and apigenin upregulated the mRNA levels of claudin-1 and ZO-1 in the ileum after exposure to DQ. These results demonstrated that resveratrol and apigenin supplementation alleviated OS via the Nrf2 signal pathway and improved intestinal barrier function in pullets exposed to DQ to a certain extent. Curcumin (CUR), a diketone substance derived from the rhizomes of Curcuma longa, is known for its various biological properties, including antioxidative, anti-inflammatory, and antiviral effects [244,245]. Research by Wu et al. (2024) revealed that the concentrations of acetate and total SCFAs in the cecum of broilers treated with CUR were notably elevated compared to those in the DQ group, suggesting that CUR facilitated acetate generation [246]. This might be because *Ruminococcaceae_Clostridium* populations in the CUR group obviously increased. Recent studies have indicated that *Ruminococcaceae* shows a crucial effect on degrading starch and fiber, thereby facilitating the generation of SCFAs [247]. As the primary component of SCFAs, acetate is directly related to the content of total SCFAs. Furthermore, the beneficial effects of CUR on preserving cecal physiological equilibrium facilitated acetate generation. Additionally, the LEfSe analysis in Wu’s study indicated that in the pairwise comparison group, *Lactobacillaceae* and *Lactobacillus* exhibited the highest LDA values. Based on this, researchers hypothesized that these two taxa might play an important role in the mitigatory effects of CUR on OS and inflammatory responses [246].

Selenium, a critical micronutrient, exhibits important effects on antioxidant, anticancer, antiviral, and immune-modulatory activities [248]. The low poisonousness and highly bioavailable properties of SeNPs make them promising candidates for therapeutic applications and selenium supplements. REG3G is closely linked to the integrity and functionality of the gut barrier. A deficiency in REG3G can promote the proliferation of mucosa-adherent bacteria, ultimately contributing to gut barrier impairment [249]. A study by Qiao et al. (2022) [95] indicated that the jejunal villi were arranged disorderly, the VH was shortened, and the number of goblet cells was reduced in mice under DQ stimulation. However, the supplementation of biogenic SeNPs in the diet increased jejunal MUC2 and REG3G expression, demonstrating their efficacy in mitigating DQ-caused gut barrier damage. Additionally, a decrease in the F/B ratio and the levels of *Desulfovibrio* and an obvious increase in the levels of *Bacteroides* and *Clostridium_XlVa* were observed in mice fed with SeNPs. *Bacteroides*, an essential component of the gut microbiota, plays a vital role in carbohydrate metabolism and propionate production [250,251]. Recent advances in microbiome research revealed that *Bacteroides thetaiotaomicron* is crucial for absorbing nutrition and contributing to barrier integrity via promoting goblet cell maturation and regulating mucus secretion [252]. *Clostridium_XlVa* is found to possess the ability to break down carbohydrates and generate SCFAs, including acetate and butyrate [253,254]. *Desulfovibrio*, an opportunistic pathogen, generates the LPS endotoxin and exhibits higher levels in the gut of obese individuals and patients with ulcerative colitis compared to healthy subjects. Furthermore, elevated levels of *Desulfovibrio* can impair IECs and the intestinal barrier integrity [255,256]. As key metabolites of the intestinal microbiota, changes in SCFA levels were consistently found in the cecum of mice following SeNP supplements. In Qiao’s other article, metabolic pathway analysis using the KEGG database suggested that the predominant metabolic pathways affected by SeNP intervention were primarily associated with aldosterone synthesis and secretion, as well as glutathione metabolism [257]. Aldosterone plays a crucial regulatory role in the sodium balance and has been demonstrated to directly contribute to end-organ injury through various pathophysiological mechanisms [258]. Glutathione, a potent tripeptide antioxidant, plays a crucial role in xenobiotic metabolism and biotransformation processes. Its reactive sulfhydryl group, primarily located on the cysteine residue, enables the effective binding and neutralization of various toxic compounds, thereby conferring significant detoxification capacity [259]. The information above offered substantial evidence supporting the critical interplay between selenium supplementation, gut microbiome, and systemic metabolic regulation.

*Periplaneta americana* L., commonly known as the American cockroach, is a medically significant insect species belonging to the genus *Periplaneta* within the family *Blattidae*. This arthropod has been traditionally utilized in Chinese medicine and is recognized as a distinctive therapeutic agent. Recent pharmacological studies have demonstrated that *Periplaneta americana* extract (PAE) exhibits significant therapeutic potential in the management of various inflammatory disorders [260,261]. Lu et al. (2022) [262] elaborated that DQ damaged gut microbiota homeostasis, characterized by elevated levels of *Bacteroidetes* and *Proteobacteria* phyla, coupled with a marked reduction in *Firmicutes* populations. This microbial dysbiosis was associated with the development of intestinal barrier dysfunction (leaky gut), as evidenced by elevated levels of gut-derived bacterial metabolite DLA in systemic circulation. Notably, PAE intervention demonstrated protective effects by ameliorating DQ-induced reduction in α-diversity indices of gut microbiota. Furthermore, PAE treatment significantly modulated microbial composition, particularly through enhancing the proliferation of the beneficial bacterium *Akkermansia muciniphila* while suppressing the growth of *Bacteroidetes* in DQ-treated mice. *Akkermansia muciniphila*, an anaerobic bacterium, was first isolated two decades ago and has emerged as a significant member of the complex microbial community in the gut [263]. Moreover, *A. muciniphila* is identified as a promising next-generation probiotic [264,265]. *A. muciniphila* has been found to upregulate the levels of intestinal TJs including occludin and ZO-1 [266]. In addition, it was also reported that *A. muciniphila* can potentially offer numerous health benefits by elevating SCFAs generation [267].

Changes in gut microbiota and metabolites observed in experimental animal models after treatment with DQ or intervention are shown in Table 1.

## 6. Conclusions and Perspective

As previously mentioned, DQ is prevalent in the environment because of extensive contamination. This results in DQ being imperceptibly ingested by humans via contaminated crops and drinking water. Therefore, it is necessary to attenuate DQ-caused OS and elevate antioxidant ability. The intervention of antioxidative medicines can be effective in eliminating or inhibiting the production of ROS and RNS, which are known to play prominent roles in DQ poisoning. As the gut serves as the primary interface between the body and its luminal environment, it is particularly susceptible to oxidative injury compared to other organs. This heightened susceptibility is due to the substantial accumulation of diet-derived oxidants, carcinogens, and mutagens within the intestinal lumen [268,269]. Based on the above information, researchers have increasingly recognized that targeting the gut microbiota and its metabolites will pave the way for innovative strategies in the prevention and intervention of OS. This article highlighted the impact of DQ toxicity on the intestinal microbiota and may act as a valuable resource for both clinical practice and fundamental investigation. By exploring the effects of DQ exposure on gut health, it aims to deepen our comprehension of the intricate relationship between environmental toxins and intestinal well-being. Nevertheless, the results of the animal experiments mentioned above are not entirely consistent. We hypothesize that these differences may arise from interspecies discrepancies. For instance, some studies were conducted in mouse models, some were performed in piglets. This hypothesis is reasonable given the well-documented physiological, metabolic, and intestinal microbial discrepancies between mice and piglets. Furthermore, host genetics, individual variations, dietary habits, age, sex, comorbidities, and environmental factors may also affect the results [270,271]. It is worth mentioning that these results acquired from animal models may not directly translate to humans. Therefore, future studies conducted in humans or in preclinical animal models are needed to reveal complex and multifactorial mechanisms responsible for gut dysbiosis establishment in order to meet the demand for personalized therapies. There is still a wide space to explore the mechanism behind the toxicity caused by DQ exposure to gut microbiota and related metabolites. Multiple strategies have been investigated in clinical trials, ranging from therapies aimed at depleting the pathogens with antimicrobials to therapies aimed at remodeling “normal” commensals with probiotics or fecal transplantation.

Notably, beyond elucidating the mechanisms of intestinal injury induced by DQ, our research team has extended the investigative scope to explore the multi-organ crosstalk mediated by the gut–brain axis, gut–kidney axis, and gut–liver axis. By establishing a cross-organ toxicity assessment model, we revealed that DQ-induced gut barrier disruption not only triggers localized inflammatory responses but also exacerbates systemic pathophysiology through distinct yet interconnected pathways such as (1) neuroinflammatory cascades in the central nervous system via circulating microparticles (gut–brain axis), (2) aggravated renal OS due to dysregulated gut microbiota-derived metabolites (gut–kidney axis), and (3) impaired hepatic detoxification capacity mediated through the portal venous system (gut–liver axis). Looking ahead, integrating organ-on-a-chip platforms with multi-omics technologies will further unravel the centrality of the “gut-X axis” in environmental-exposure-related pathologies, thereby advancing toxicology into a systems biology era focused on holistic mechanistic decoding.

## Figures and Tables

**Figure 1 antioxidants-14-00721-f001:**
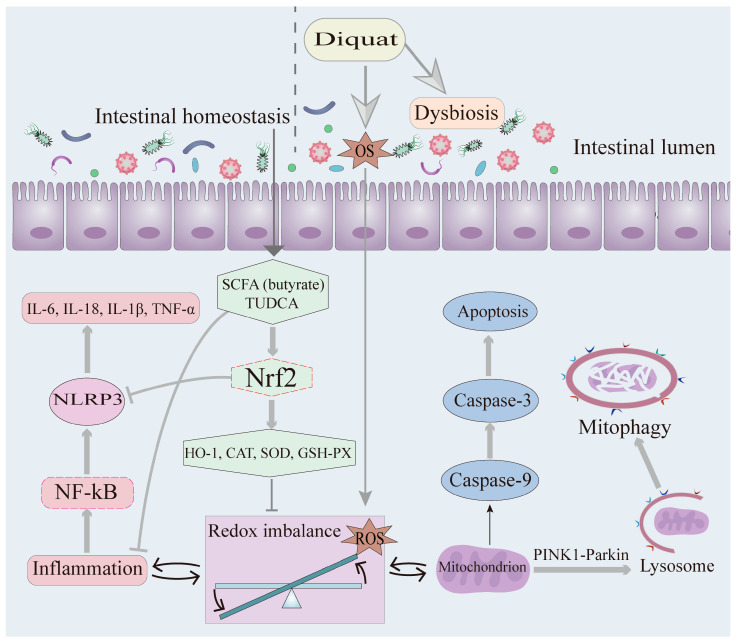
Schematic overview illustrating the multiple pathways through which diquat triggers ROS production, inflammatory factor release, and the activation of mitophagy and apoptosis in IECs. (1) SCFAs (e.g., butyrate) and TUDCA derived from intestinal homeostasis activate Nrf2 signaling to upregulate antioxidant enzymes (HO-1/CAT/SOD/GSH-PX), constituting a critical defense against oxidative injury. (2) DQ elicits OS through dual mechanisms: the direct induction of ROS generation and the indirect disruption of redox homeostasis via dysbiosis. (3) Pro-inflammatory signaling: NF-κB/NLRP3 activation upregulates cytokines (IL-6, IL-1β, IL-18, TNF-α). (4) Mitochondrial–lysosomal axis dysfunction: ROS accumulation induces PINK1-Parkin-dependent mitophagy, and lysosomal activity mediates the clearance of damaged mitochondria. (5) OS triggers mitochondrial-mediated apoptosis (caspase-9/caspase-3 cascade activation). SCFA: Short-chain fatty acid; TUDCA: Tauroursodeoxycholic acid; Nrf2: Nuclear factor erythroid 2-related factor 2; HO-1: Heme oxygenase-1; CAT: Catalase; SOD: Superoxide dismutase; GSH-PX: Glutathione peroxidase; ROS: Reactive oxygen species; NF-kB: Nuclear factor kappa-B; NLRP3: NLR family pyrin domain containing 3.

**Figure 2 antioxidants-14-00721-f002:**
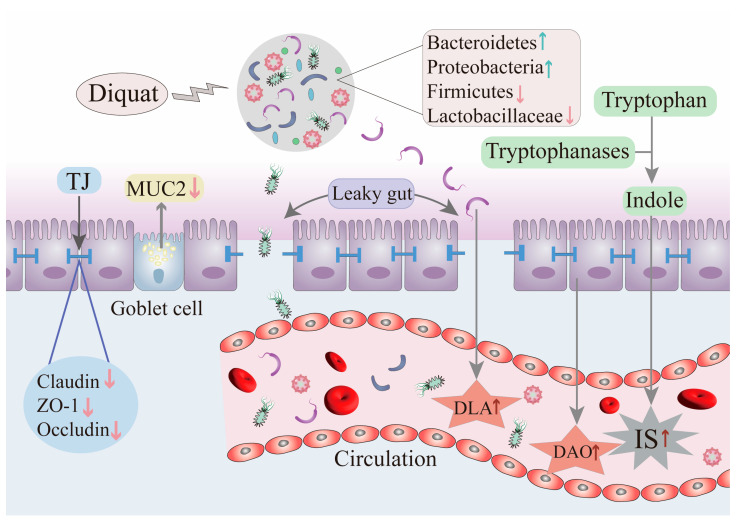
DQ compromises the intestinal barrier and increases intestinal permeability. (1) Tight junction (TJ) disruption: the downregulation of claudin, ZO-1, and occludin compromises epithelial barrier integrity. (2) Reduced MUC2 secretion from goblet cells exacerbates mucosal vulnerability. (3) DQ elevates serum DAO and DLA levels, verifying the impairment of gut barrier integrity. (4) DQ enhances tryptophan metabolism, thereby augmenting the production of IS, a uremic toxin associated with renal pathophysiology. ↑: upregulation; ↓: downregulation; TJ: Tight junction; ZO-1: Zonula occludens-1; MUC2: Mucin 2; DLA: D-lactate; DAO: Diamine oxidase; IS: Indoxyl sulfate.

**Table 1 antioxidants-14-00721-t001:** Changes in gut microbiota and metabolites of experimental animal models after treatment with DQ or intervention.

Subjects	Changes (DQ)	Intervention	Changes (Intervention)	Ref.
C57BL/6 mice	*Allobaculum*, *Providencia*, *Escherichia-Shigella*, *Bacteroidetes*, *proteobacteria*↑; *Firmicutes*↓	*Periplaneta americana* extract (PAE)	*Akkermansia muciniphila*↑; *Bacteroidetes*↓	[262]
Weaned piglets	HCA, HDCA, TUDCA↓	Hydroxytyrosol	HCA, HDCA, TUDCA↑	[137]
ICR mice	*Firmicutes*↓; *Bacteroidetes*↑	Hydroxytyrosol	*Firmicutes*, *Lactobacillus*↑; *Bacteroidetes*↓; butyrate↑; glycerophospholipid metabolism, pentose, glucuronate interconversions↓	[173]
C57BL/6J mice	*Bacteroidota*, *Coriobacteriia*, *Enterorhabdus*↓; *Escherichia–Shigella*↑	*L. delbrueckii*, *L. amylovorus*, and *L. salivarius* EVs	*Enterococcus*↓; *Parasutterella*, *Bifidobacterium*, *Erysipelatoclostridium*↑	[230]
Mice	*Firmicutes*↓	Eucommia ulmoides polysaccharide (EUPS)	*Firmicutes*, *Ligilactobacillus*↑; *Helicobacter*↓	[168]
WOD168 broilers	*Lactobacillus*, *Alistipes*↓	Quercetagetin (QG)	*Lactobacillus*, *Alistipes*↑	[178]
Weaned piglets	*Firmicutes*, *Actinobacteria*, *Ruminococcaceae* UCG-005, *Eubacterium coprostanoligenes*↑	Resveratrol (RES)	*Clostridium sensu stricto* 1, *Lachnospiraceae*↑	[171]
C57BL/6 mice		Selenium nanoparticles (SeNPs)	*Akkermansia*, *Muribaculaceae*, *Bacteroides*, *Parabacteroides*↑	[257]
C57BL/6 mice	*Bacteroides*, *Helicobacter*↑; *Firmicutes*, *Pediococcus*, *Enterococcus*, *Dubosiella*↓	*Pediococcus pentosaceus* ZJUAF-4; VC	ZJUAF-4 reversed these changes induced by DQ and the reversed abilities of ZJUAF-4 seemed to be higher than those of VC	[63]
Cobb broilers	*Lactobacillaceae*, *Victivallis*, *Bacillus*↓; *Saccharopolyspora_hir-suta*, *Staphylococcus_succinus*↑	Curcumin (CUR)	*Lactobacillaceae*, *Ruminococcaceae_Clostridium*↑; *Saccharopolyspora_hirsuta*, *Staphylococcus_succinus*↓; acetate, total SCFAs↑	[246]
C57BL/6J mice		Hydrogenobacter thermophilus strain (HtSOD)	*Dubosiella*, *Alistipes*↑	[42]
Weaned piglets	*Acidobacteria*↓; *Turicibacteraceae*, *Clostridium*, *Turicibacter*↑	*Bacillus* amyloliquefaciens SC06	*Ruminococcaceae, Clostridium*↓; *Pasteurellaceae*, *Lactobacillus*, *Actinobacillus*↑	[81]
SD rats	*g_Lactobacillus, p_Firmicutes*, *g_Akkermansia*, *p_Verrucomicrobia*↓; *p_Proteobacteria*, *g_Escherichia*, *s_Escherichiacoli*↑	*Bacillus* SC06	*g_Anaerofilum*, *s_Bacteroides uniformis*↑; *s_Oscillospira guilliermondil*↓	[165]
C57BL/6 mice		SeNPs	*Bacteroidetes*, *Clostridium_XlVa*↑; *Verrucomicrobia*, *Desulfovibrio*↓; total SCFAs, butyrate, isobutyrate, valerate, isovalerate↑	[95]

↑: upregulation; ↓: downregulation; DQ: Diquat; Ref.: References; HCA: Hyocholic acid; HDCA: Hyodeoxycholic acid; TUDCA: Tauroursodeoxycholic acid; EVs: Extracellular vesicles; SCFAs: Short-chain fatty acids.

## Data Availability

Data sharing is not applicable to this article as no new data were created or analyzed in this study.

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
