# Peer review of "Impact of Diquat on the Intestinal Health and the Composition and Function of the Gut Microbiome"

_antioxidants, 2025, doi:10.3390/antiox14060721_

Round 1
Reviewer 1 Report
The review entitled "Impact of diquat on the composition and function of the gut microbiome" is very interesting, sound, clear and well carried out. I would only suggest 3 minor changes.
1) Broader title - Suggestion: "Impact of diquat on intestinal health and the composition and function of the gut microbiome"
2) Better explain the PQ poisoing in humans (i.e. doses):
| Dose (mg/kg body weight) | Estimated Ingested Volume | Expected Outcome |
|---|---|---|
| < 20 mg/kg | ~5–10 mL of 20% solution | Mild poisoning: full recovery likely with supportive care |
| 20–40 mg/kg | ~10–20 mL of 20% solution | Moderate poisoning: kidney/liver/lung damage, survival possible |
| > 40 mg/kg | >20 mL of 20% solution | Severe poisoning: almost always fatal due to multi-organ failure |
-
Lethal dose (LD₅₀) for humans is estimated at ~35 mg/kg orally.
-
Ingestion of >30 mL of a 20% solution (i.e., Gramoxone) is usually fatal without immediate and aggressive treatment.
3) It could be interesting to describe the PQ poisoning effects on gut-brain axis
See above
Reviewer 2 Report
The paper is very interesting and it open new therapeutic options for the future. Athough I suggest:
- At page 10, line 391, the Authors should precise the different types of bacteria classified within Firmicutes Phylum
- At page 10, line 400., the Authors should select the bacteria priducing tryptophan
The paper is very interesting and it open new therapeutic options for the future. Athough I suggest:
- At page 10, line 391, the Authors should precise the different types of bacteria classified within Firmicutes Phylum
- At page 10, line 400., the Authors should select the bacteria priducing tryptophan
Reviewer 3 Report
In section 2.1 Authors refer only to Oxidative stress while in the next section 2.2 they refer to DQ effects on oxidative stress and inflammation. Thus, the authors should insert inflammation in the 2.1 section or include the 2.1 content in 2.2 section.
Lines 198 – 223 refers to SCFA and oxidative stress, which the relation in the mitophagy section?
Fig.1: Legend should be more exhaustive.
Fig.1: In this figure authors refer to apoptosis. There are evidence concerning the DQ effects on apoptosis, mostly at intestinal level?
Fig.2: The effect od DQ in this sequence is too speculative. Many factors are not mentioned (e.g tryptophan, thryptophanase, indole), goblet cells are mentioned only pages later. Evidences should support this sequence and the figure should be organized at the end of the paragraph 2.4.
Abbreviations should be indicated (e.g VH and CD)
The lines from 396 to 420 is not clear and it should better supported by literature, mostly concerning IS modulation in CKD and the possible DQ effect.
Many aspects/descriptions, mostly on ROS, are repeated several times.
In section 2.1 Authors refer only to Oxidative stress while in the next section 2.2 they refer to DQ effects on oxidative stress and inflammation. Thus, the authors should insert inflammation in the 2.1 section or include the 2.1 content in 2.2 section.
Lines 198 – 223 refers to SCFA and oxidative stress, which the relation in the mitophagy section?
Fig.1: Legend should be more exhaustive.
Fig.1: In this figure authors refer to apoptosis. There are evidence concerning the DQ effects on apoptosis, mostly at intestinal level?
Fig.2: The effect od DQ in this sequence is too speculative. Many factors are not mentioned (e.g tryptophan, thryptophanase, indole), goblet cells are mentioned only pages later. Evidences should support this sequence and the figure should be organized at the end of the paragraph 2.4.
The lines from 396 to 420 is not clear and it should better supported by literature, mostly concerning IS modulation in CKD and the possible DQ effect.
Many aspects/descriptions, mostly on ROS, are repeated several times.
Round 2
Reviewer 1 Report
The authors have siginficantly improved the paper. Congratulations
None
Author Response
Thank you for your encouraging comments! We are truly pleased that the improvements meet your approval. Your expertise was invaluable to this work.
Reviewer 2 Report
The Authors have inserted and completed the suggestions of the reviewers. I think that the paper can be published in the journal
The Authors have inserted and completed the suggestions of the reviewers. I think that the paper can be published in the journal
Author Response

(The authors gave the same response as above.)

Reviewer 3 Report
Comments 2: [Lines 198 – 223 refers to SCFA and oxidative stress, which the relation in the mitophagy section?]
Response 2: At the end of the paragraph (page 5, lines 214-217), we mentioned the
relationship among SCFA, oxidative stress and mitophagy:
The data of Wang et al. (2019) indicated that DQ injection caused serious intestinal OS in
pigs, while butyrate relieved the intestinal OS and inflammation, and improved
mitochondrial function through selectively inducing mitophagy.
This answer needs to be expanded by adding more details in the text
Comments 2: [Lines 198 – 223 refers to SCFA and oxidative stress, which the relation in the mitophagy section?]
Response 2: At the end of the paragraph (page 5, lines 214-217), we mentioned the
relationship among SCFA, oxidative stress and mitophagy:
The data of Wang et al. (2019) indicated that DQ injection caused serious intestinal OS in
pigs, while butyrate relieved the intestinal OS and inflammation, and improved
mitochondrial function through selectively inducing mitophagy.
This answer needs to be expanded by adding more details in the text
